# Phase-Controllable Chemical Vapor Deposition Synthesis of Atomically Thin MoTe_2_

**DOI:** 10.3390/nano12234133

**Published:** 2022-11-23

**Authors:** Tao Xu, Aolin Li, Shanshan Wang, Yinlong Tan, Xiang’ai Cheng

**Affiliations:** 1College of Advanced Interdisciplinary Studies, National University of Defense Technology, Changsha 410073, China; 2Powder Metallurgy Research Institute, Central South University, Changsha 410073, China; 3Science and Technology on Advanced Ceramic Fibers and Composites Laboratory, College of Aerospace Science and Engineering, National University of Defense Technology, Changsha 410073, China

**Keywords:** 2D MoTe_2_, space-confined CVD, phase-controllable synthesis, Te vacancy, tellurization velocity, lattice distortion

## Abstract

Two-dimensional (2D) molybdenum telluride (MoTe_2_) is attracting increasing attention for its potential applications in electronic, optoelectronic, photonic and catalytic fields, owing to the unique band structures of both stable 2H phase and 1T′ phase. However, the direct growth of high-quality atomically thin MoTe_2_ with the controllable proportion of 2H and 1T′ phase seems hard due to easy phase transformation since the potential barrier between the two phases is extremely small. Herein, we report a strategy of the phase-controllable chemical vapor deposition (CVD) synthesis for few-layer (<3 layer) MoTe_2_. Besides, a new understanding of the phase-controllable growth mechanism is presented based on a combination of experimental results and DFT calculations. The lattice distortion caused by Te vacancies or structural strain might make 1T′-MoTe_2_ more stable. The conditions for 2H to 1T′ phase conversion are determined to be the following: Te monovacancies exceeding 4% or Te divacancies exceeding 8%, or lattice strain beyond 6%. In contrast, sufficient Te supply and appropriate tellurization velocity are essential to obtaining the prevailing 2H-MoTe_2_. Our work provides a novel perspective on the preparation of 2D transition metal chalcogenides (TMDs) with the controllable proportion of 2H and 1T′ phase and paves the way to their subsequent potential application of these hybrid phases.

## 1. Introduction

Distinct from zero-gapped graphene [1,2], the increasingly popular two-dimensional (2D) transition metal chalcogenides (TMDs) have attracted research interests in the electronic, optoelectronic, photonic and catalytic fields due to their tunable bandgaps, polymorphism [3,4,5,6,7,8]. As a unique kind of TMDs, 2D molybdenum ditelluride (MoTe_2_) has two stable phases: semiconducting 2H phase and metallic 1T′ phase. 2H MoTe_2_ is a hexagonal crystal with a direct energy bandgap of 1.1 eV for a single layer and with an indirect energy bandgap of 1.0 eV for bulk MoTe_2_ [9]. This bandgap corresponds to the infrared region and is very close to that of Si (1.1 eV), revealing applications in silicon-integrated optoelectronic transistors, field effect transistors (FET), and photonic devices [10,11,12,13,14,15]. 1T′ MoTe_2_ exhibits semi-metallic characteristics with large magnetoresistance, superconductivity and exotic Weyl state, and possesses strong spin-orbit coupling effects, which can be employed in quantum computing and spintronics [16,17,18,19,20]. Notably, the potential barrier between 2H and 1T′ MoTe_2_ is smaller [21,22,23,24] than that of other group VI TMDs, such as MoS_2_, MoSe_2_, WS_2_, and WSe_2_, which makes it easier for the phase transition. Hence, the stable 2H-MoTe_2_ and 1T′-MoTe_2_ coplanar heterostructures can be prepared to reduce the contact resistance and enhance the electron mobility in the phase-engineering application [25,26,27]. To realize the applications mentioned above for different phases of 2D MoTe_2_, a thorough knowledge of the phase transformation mechanism and the strategies for obtaining a specific phase in a controlled manner is highly desirable.

To date, both theoretical and experimental studies were conducted to address the phase transition between 2H and 1T′ MoTe_2_. The DFT calculation results announced that laser irradiation [28,29], mechanical strain [30,31], adsorbed atoms and molecules [22], and doped charges [32] could induce conversion from 2H to 1T′ MoTe_2_. And some experiments succeeded in achieving phase transition by the external incentives referred to above [24,33,34,35]. However, the external incentives were just localized and only small-area phase conversion was achieved. And these post-treating methods also increase the complexity of the process and result in poor controllability. Considering the poor operability of these methods, researchers are looking for simpler and more efficient methods to achieve phase-controllable synthesis in one step. The CVD growth method is highly productive and convenient in the preparation of 2D materials, but direct CVD synthesis of atomically thin MoTe_2_ domains with controllable phases is still challenging. The 1T′ and 2H MoTe_2_ synthesized through the CVD methods previously reported is relatively thick (>three layers) and polycrystalline in general because of the thick and poor- crystallization precursor. The controllable synthesis of few-layer (<three layers) 1T′ and 2H MoTe_2_ single grains with high quality is needed for their ideal performance related to high specific surface area and abound edge atoms.

In addition, it remains ambiguous how the 2H- and 1T′-MoTe_2_ grow and convert each other. Among some investigations conducted on the controllable CVD synthesis of 2H and 1T′ MoTe_2_ [26,36,37,38,39,40,41,42], some primary progress has been made indeed. 2H-MoTe_2_ dominates when MoO_3_ film is used as the precursor and conversely 1T′ phase prevails when using the pure Mo or MoO_x_ (x < 3) film. This correspondence between reactants and products might be attributed to the different structure strain originating from the lattice adjustment during the transformation from Mo precursor to MoTe_2_ [39]. Besides, Te deficiency could stabilize 1T′ MoTe_2_ while 2H MoTe_2_ is derived at ample Te supply, and 2H-1T′ MoTe_2_ homojunction was synthesized at a moderate Te concentration [40]. Moreover, Park et al. [41] reported that excess Te might trigger the 1T′-to-2H phase transition and stabilize the 2H phase by relaxing the lattice strain while another study found that excessive Te adsorbed on the surface of MoTe_2_ is conducive to the formation of 1T′ phase [42]. Thus, the theory about the influence of Te concentration on the phase transition needs more experimental verification and remains to be developed.

In this work, we realize the large-scale synthesis of atomically thin MoTe_2_ domains with a controllable phase ratio of 2H to 1T′ by regulating the carrier gas flow rate, the concentration of Na_2_MoO_4_ solution and the heating temperature of Te powder. To unveil the effect of the experimental conditions above, density functional theory (DFT) calculations are performed. Based on the experimental and simulated results, a growth mechanism involving the phase transition process between 2H and 1T′ MoTe_2_ is proposed. 1T′-MoTe_2_ domains become prevailing because of lattice distortion resulting from insufficient Te source or the fast tellurization velocity. The lack of Te supply may easily cause Te vacancies in produced MoTe_2_ and the fast tellurization process induced lattice strain, both of which could result in lattice distortion. In contrast, the dominating 2H-MoTe_2_ should be prepared with sufficient Te supply and under a slow tellurization process, simultaneously. Our work provides a new perspective for the phase-controllable synthesis of few-layer 2H and 1T′ TMDs and promotes their potential applications.

## 2. Materials and Methods

### 2.1. Synthesis of Atomically Thin 2H-MoTe_2_ and 1T′-MoTe_2_

The growth of atomically thin MoTe_2_ was performed with a single-zone furnace (SK−G03123K, Tianjin Zhonghuan, Tianjin, China), in which the space-confined CVD setup consisting of two stacked silicon slices (280 nm SiO_2_/Si, 1.7 cm × 1 cm) was adopted. Sodium molybdate (Na_2_MoO_4_, 99%, Aladdin) and tellurium powder (Te, 99.99%, Aladdin) were used as the Mo and Te precursor, respectively. To make the SiO_2_/Si substrate more hydrophilic, it was pretreated with oxygen plasma for 50 s. Then 10 μL of 0.01 mol/L Na_2_MoO_4_ aqueous solution was spin-coated on the SiO_2_/Si substrate, which was the bottom one. Another bare SiO_2_/Si substrate was stacked up on it in a face-to-face way, constructing a narrow gap of 24 μm (Appendix A). This pair of stacked slices was initially placed downstream away from the heated zone. A total of 80 mg Te powder was put in an alumina (Al_2_O_3_) crucible was placed at the upstream, 6.2 cm away from the furnace’s heating-zone edge in the upstream direction to ensure the Te powder could vaporize. Before the reaction, a high flow of argon (Ar, 500 sccm, regulated by the mass flow meter ZL−03H, Tianjin Zhonghuan, Tianjin, China) was used to flush the CVD system for 10 min. Then the heating center was heated up to 650 °C by 20 °C/min under 10 sccm Ar. Next, Te powder was subsequently preheated for 5 min under a mixed carrier gas of H_2_/Ar (10 vol% H_2_) at a flow rate of 70 sccm. And then, the substrates for growth were moved to the heated center by the magnet where they remained for 20 min under the constant flow rate of H_2_/Ar. Finally, the reaction procedure was terminated by fast cooling of the furnace and simultaneously 500 sccm Ar flow was applied to flush the CVD system for 10 min. As a result, the atomically thin MoTe_2_ was obtained at the upper SiO_2_/Si substrate. To investigate the influence of gas flow rate on phase state, it was set as 10, 20, 40, 60, 70, 80, 100 and 120 sccm gradually, while the heating temperature was at 650 °C. To explore the impact of the concentration of Na_2_MoO_4_ solution, 0.005, 0.0075, 0.01, 0.0125, 0.015 and 0.0175 mol/L solution were prepared and 650 °C was also applied. As for the study on the influence of heating temperature of Te powder, the heating temperature of the single-zone furnace was set at 800 °C to make the concentration of Te vapor sufficient and avoid the effect of Te vacancies on the phase state. Thus, the heating temperature of Te was regulated by varying its location at L (5.3, 6.2, 7.2, and 7.8 cm) from the upstream end.

### 2.2. Characterization

The phase of few-layer 2H-MoTe_2_ and 1T′-MoTe_2_ was characterized by optical microscopy (Nikon ECLIPSE LV150NL, Tokyo, Japan), scanning electron microscope (Hitachi-SU8010, Tokyo, Japan) and Raman spectroscopy (WITec, Alpha 300R, Ulm, Germany) with an excitation laser of 532 nm. These statistics of the ratio for 2H- and 1T′-MoTe_2_ were analyzed by the software ImageJ (v1.8.0, from National Institutes of Health, Bethesda, MD, USA) and only the domains larger than 2 μm can be counted. The samples analyzed for each variable were taken from the representative optical images of three experiments under the same synthetic condition. The atomic force spectroscopy (Bruker Icon, Billerica, MA, USA) was used to measure the thickness of as-grown MoTe_2_.

### 2.3. Calculation Method

DFT calculations were performed using the generalized gradient approximation (GGA) in the Perdew-Burke-Ernzerhof (PBE) form for the exchange and correlation potential [43], as implemented in the Vienna Ab Initio Simulation package (VASP) [44]. The energy cutoff of the plane wave basis was set to 500 eV. Monkhorst-Pack k-meshes were adopted to sample the first Brillouin zone with a k-point density of no less than 20.0/Å^−1^. We used unit cells to simulate the strain effects of defect-free MoTe_2_ monolayers and used supercells to simulate the monolayer MoTe_2_ with point defects, which contain 4 to 24 unit cells depending on the concentration of vacancy (see Appendix A for details). The in-plane lattice constants and all the atomic positions were relaxed. The energy and force convergence conditions for structural optimization were set to 10^−5^ eV/f.u. and 0.01 eV/Å, respectively. A vacuum space larger than 15 Å was used to screen the false interactions induced by the periodic boundary conditions.

## 3. Results and Discussion

To make few-layer MoTe_2_ domains more homogenous, a space-confined CVD system was applied as reported in our previous work [45], in which fluid dynamics are equivalent to the low-pressure CVD (LPCVD). Figure 1a shows the schematic illustration of this space-confined CVD system possessing a narrow gap of 24 μm. The substrates were put in the heating center of the single-zone furnace while Te powder was placed at the upstream during the reaction process (see Section 2.1). The synthesis was carried out at 650 °C by atmospheric pressure CVD (APCVD) with the mixed carrier gas of hydrogen and argon. To ensure the Te source was well evaporated and then diffused to the reaction zone, the Te powder was preheated for 5 min before the Na_2_MoO_4_ spined substrate was pushed to the reaction center. The temperature-elevating profile of Te powder and Na_2_MoO_4_ is shown in Figure 1b.

Figure 1c shows that the as-grown grains possess leaf-like morphology with a lateral size of 40–80 μm and uniform thickness, which is demonstrated to be 1T′-MoTe_2_ in Figure 2a. They were obtained when the gas flow rate is 40 sccm H_2_/Ar and the concentration of Na_2_MoO_4_ is 0.01 mol/L. Since 1T′-MoTe_2_ has a monoclinic crystal structure (space group *P*2_1_/*m*) [46], its single crystal morphology is mostly leaf-like and elongated [27]. The height difference of the single leaf along the red dotted line is near 1.98 nm (Figure 1d), illustrating the number of layers of 1T′-MoTe_2_ is ~2 [47,48]. In contrast, crystals with hexagonal morphology and uniform size are verified to be 2H-MoTe_2_ by the following Raman spectrum (Figure 2b) and correspond to the 2H phase of the hexagonal crystal system (space group *P*6_3_*/mmc*) [27,49], which are synthesized under the condition of 70 sccm H_2_/Ar and 0.01 mol/L Na_2_MoO_4_. The AFM image of the hexagonal domain (Figure 1f) exhibits that the thickness of the 2H-MoTe_2_ crystal is ~1.91 nm, approximately corresponding to that of a bilayer [50].

For the 1T′-MoTe_2_ grains, Figure 2a shows the Raman-active modes of A_g_ (86 cm^−1^, 112 cm^−1^, 127 cm^−1^, 252 cm^−1^, 269 cm^−1^) and B_g_ (102 cm^−1^, 162 cm^−1^, 185 cm^−1^), which is consistent with previous reports on 1T′-MoTe_2_ [45]. The Raman mapping of the most prominent B_g_ (162 cm^−1^) peak of 1T′-MoTe_2_ presents a uniform intensity distribution within the leaf-like shape (the inset of Figure 2a), which displays a highly homogenous 1T′-MoTe_2_ grain [47,51,52,53,54]. The out-plane characteristic peak of the A_g_ mode located at 269 cm^−1^ that is sensitive to the thickness of the material, indicates that the grain characterized here is monolayer [23]. The AFM image for monolayer 1T′-MoTe_2_ (Appendix A) demonstrates a rough surface with discrete particles because of the fast degradation when exposed to air. For the 2H-MoTe_2_ grains, the Raman-active modes of A_1g_ (~172 cm^−1^) and B_2g_^1^ (~289 cm^−1^) are out-of-plane modes whereas E_2g_^1^ (~234 cm^−1^) is an in-plane mode [10,23,36,55]. The inset in Figure 2b displays the Raman mapping for the representative E_2g_^1^ of the 2H-MoTe_2_ domain, revealing a uniform and defect-free crystal grain of hexagon. The profile of 1T′-MoTe_2_ and 2H-MoTe_2_ grains in the Raman mapping is consistent with that of optical images. Combining the AFM images with the Raman data, it could be confirmed that the majority of products are single-layer and bilayer MoTe_2_ crystals. Moreover, the in situ homojunction between 1T′ and 2H can also be formed in our CVD system (Appendix A), which can be applied in the phase-engineering devices for ohmic contact.

### 3.1. Synthesis of Few-Layer MoTe_2_ with Different Phase Proportions by Controlling the Gas Flow Rate

Because Te vapor was delivered by carrier gas, the concentration of Te was proportional to the gas flow rate. Considering the Te content plays an important role in the phase of MoTe_2_ reported in the previous study [36,40,41,56], the impact of gas flow rate on phase transition was investigated in our experiments. First, the H_2_/Ar gas flow rates of 10, 20, 40, 60, 70, 80, 100 and 120 sccm were applied at 650 °C. Grains with different morphologies and phases were obtained (see Figure 3a–d and Appendix A). At the relatively low (10, 20 sccm) and high (100, 120 sccm) gas flow rates, the resulting MoTe_2_ grains are mainly leaf-like while domains of leaf and hexagon coexist at the moderate gas flow rate (40–80 sccm). Given the correspondence between the crystal morphology and phase that was deduced from the Raman mapping results (Figure 2) and statistics for phase states of grains with different shapes (e.g., 100% 2H-MoTe_2_ for 20 hexagonal grains and 100% 1T′-MoTe_2_ for 20 leaf-like domains, whose Raman spectra are presented in Appendix A), we have the primary assumptions that the hexagon grains are 2H-MoTe_2_ and the leaf-like crystals are 1T′-MoTe_2_. The ratio of the grain quantity and area for 2H-MoTe_2_ as a function of the gas flow rate is summarized in Figure 3e. It is clear that the ratio of 2H-MoTe_2_ increases first and then decreases with the largest proportion at 70 sccm, both for the quantity and area ratio. As Appendix A demonstrated, the nucleation density increases accompanied by the decreasing average crystal size as the carrier gas flow rate rises, which reveals that more flux of Te source is transmitted into the confined space of two Si/SiO_2_ substrates. Namely, the flux of Te in the confined gap is positively correlated with the carrier gas flow rate. Thus, 1T′-MoTe_2_ grains dominated at a low gas flow rate (10, 20, 40 sccm) because of Te deficiency [24,26]. More 2H-MoTe_2_ domains were synthesized when the increasing Te flux was introduced by a higher gas flow rate (60, 70 sccm). The largest ratio of 2H phases was obtained at 70 sccm H_2_/Ar since the relative content of Te to Mo is almost sufficient under this condition. Surprisingly, the ratio of 2H-MoTe_2_ reduces as the flow rate further increases to 80–120 sccm. This could be attributed to the faster tellurization process under more Te vapor. Fast tellurization velocity might stabilize the 1T′ phase, thus almost 100% 1T′-MoTe_2_ was fabricated at a high gas flow rate (100, 120 sccm).

### 3.2. Synthesis of Few-Layer MoTe_2_ with Different Phase Proportions by Controlling the Concentration of Na_2_MoO_4_

On the other hand, the concentration of the Na_2_MoO_4_ solution can also influence the phase of as-grown MoTe_2_. Figure 4a–f presents the optical microscopic images of the products fabricated at varied concentrations of the Na_2_MoO_4_ solution. Based on these samples, the statistical data on the quantity ratio of 1T′/2H nucleation, the crystal size and nucleation density are displayed in Figure 4g and Appendix A. As the concentration of Na_2_MoO_4_ solution is increased, the quantity percent of 2H-MoTe_2_ firstly increases and then decreases, with the highest ratio of 86% at the Na_2_MoO_4_ concentration of 0.0075 mol/L. In contrast, we could obtain 100% 1T′-MoTe_2_ at a relatively high concentration of the Na_2_MoO_4_ solution beyond 0.015 mol/L. There are some nano-particles in Figure 4f which are proven to be the non-reacted particles related to Na_2_MoO_4_ (Appendix A) due to the lack of Te.

### 3.3. Synthesis of Few-Layer MoTe_2_ with Different Phase Proportions by Controlling the Heating Temperature of Te

In addition, we have statistically investigated the influence of the heating temperature of Te powder (*T*_Te_) on growth behavior. According to the heating temperature points for Te powder at different positions of the furnace marked in Figure 5a–d, the elevation of *T*_Te_ could result in more Te vapor and therefore faster tellurization velocity when the location of Te powder, L (illustrated in Figure 1a), varies from 5.3 cm to 7.8 cm. By changing the heating temperature of Te powder, the systematic evolution of morphology and phase of MoTe_2_ is observed in Figure 5. Typical optical images (Figure 5a–d) show the resulting few-layer MoTe_2_ crystals on the top SiO_2_/Si substrate, which have hexagonal or leaf-like shapes. Figure 5e shows that the percentage of 2H-MoTe_2_ decreases as the *T*_Te_ is enhanced. The highest percentage of 2H-MoTe_2_ in the quantity is 82% when Te powder was heated at 460 °C (5.3 cm) in the furnace. Notably, a higher concentration of Te vapor can make the tellurization velocity faster and help stabilize 1T′ phase when the location of Te is closer to the heating center (larger L).

According to our experimental results above, it is found that 1T′-MoTe_2_ might be a defect-related phase and prepared at insufficient Te supply or fast tellurization velocity, while the 2H-MoTe_2_ domains might be defect-free. This is because Te deficiency can result in a large number of vacancies in the reaction system and fast tellurization velocity might generate the accumulated structural strain during the growth of MoTe_2_. Both Te vacancies and strain can distort the lattice and make 1T′-MoTe_2_ dominate. To determine the critical conditions for the phase transition, we used the DFT calculations to study the energy difference between the 2H- and 1T′-phase of monolayer MoTe_2_ influenced by this lattice distortion. Firstly, the effect of lattice expansion induced by the rapid tellurization process was analyzed. As displayed in Figure 6a, the energy difference between 2H- and 1T′-MoTe_2_ monolayer monotonously increases as the uniaxial (*b*-axis) strain is enhanced, suggesting tensile strain along the *b*-axis can stabilize the 1T′-phase. The critical strain along the *b*-axis for the phase transition between 2H and 1T′ is ~6%. The reason why we just take the lattice strain along the *b*-axis into consideration is that the phase transition between 2H and 1T′ gets easier under uniaxial strain than biaxial strain, and the strain along the *a*-axis is less sensitive to the phase transition than the *b*-axis [30]. Moreover, we analyzed the role of Te vacancies in the phase transition. Figure 6b displays the total energy difference as a function of Te vacancy concentration. We find that when Te monovacancy concentration exceeds 4% or Te divacancy concentration exceeds 8%, the ground state of monolayer MoTe_2_ could be changed from the 2H phase to the 1T′ phase.

Given the experimental results and theoretical calculations, we propose a phase-transition mechanism as Figure 7 shows: (1) the 2H-MoTe_2_ dominates in the condition of slow tellurization velocity and sufficient Te; (2) the 1T′-MoTe_2_ prevails due to the lattice distortion triggered by Te vacancies or structural strain when the relative content of Te to Mo precursor is not enough or the tellurization velocity is fast. Figure 6b shows that a large quantity of Te vacancies is beneficial to stabilize the metallic phase, 1T′-MoTe_2_. In our experiments, both low gas flow rate (10, 20 sccm) and high concentration of the Na_2_MoO_4_ solution (0.015, 0.0175 mol/L) mean the relative concentration of Te to Mo is low, resulting in 100% 1T′-MoTe_2_. The relatively low concentration of Te to Mo implies the quantities of reactive Te atoms may be smaller than twice that of reactive Mo atoms. Under this condition, the producing Te monovacancy or Te divacancy concentration may be higher than 4% or 8%, respectively, according to our DFT results. On the other hand, the necessary condition for the formation of 2H MoTe_2_ is a sufficient Te supply, for which Te vacancies can be scarcely formed. In our case, the quantity percent of 2H-MoTe_2_ is the largest when the carrier gas flow rate is medium as 70 sccm, or the concentration of the Na_2_MoO_4_ solution is moderate (0.0075 mol/L), where there is sufficient Te relative to Mo precursor and the tellurization velocity is not fast. However, even if Te is sufficient, excessive Te relative to Mo accelerates the tellurization velocity, resulting in strain or defects greater than the phase transition critical value. Thus, it makes the 1T′ phase dominate. For example, all of the synthesized grains are 1T′-MoTe_2_ at a high carrier gas flow rate (100, 120 sccm), low concentration of the Na_2_MoO_4_ solution (0.005 mol/L) or high *T*_Te_ (580, 640 °C). In general, 100% 1T′-MoTe_2_ can easily be obtained while 100% 2H-MoTe_2_ can hardly be synthesized since the condition for synthesizing 2H-MoTe_2_ is relatively harsh, which can be reached at sufficient Te vapor and a precisely controlled tellurization velocity. Therefore, how to reduce the lattice distortion caused by Te vacancies and strain is the key to synthesizing 100% 2H-MoTe_2_.

## 4. Conclusions

To sum up, we synthesized uniform and atomically thin (<3 layers) 2H-MoTe_2_ and 1T′-MoTe_2_ single domains by the space-confined CVD system. A novel growth mechanism for 2H- and 1T′-MoTe_2_ controlled by the relative concentration of Te to Mo and tellurization velocity is proposed. 1T′-MoTe_2_ dominates due to a large quantity of Te vacancies at a relatively low concentration of Te to Mo, decided by a low gas flow rate (10–40 sccm) or high concentration of Na_2_MoO_4_ (0.01–0.0175 mol/L). 1T′-MoTe_2_ also prevails through the fast tellurization velocity because of the lattice strain, even though there is sufficient Te at a relatively high concentration of Te to Mo, decided by high gas flow rate (80–120 sccm), low concentration of Na_2_MoO_4_ (0.005 mol/L) or high *T*_Te_ (≥520 °C). DFT calculations indicate that Te monovacancy concentration exceeding 4% or Te divacancy concentration exceeding 8% or the tensile strain along the *b*-axis larger than 6% facilitates the stabilization of 1T′-MoTe_2_. The dominant 2H MoTe_2_ should be prepared under the conditions of sufficient Te content and low tellurization velocity, which is crucial to avoid the lattice distortion induced by Te vacancies and strain. This study provides a new method for phase-controllable synthesis of atomically thin TMDCs with semiconducting and metallic phases and promotes the understanding of the phase transformation mechanism, which paves the way for phase-engineering applications.

## Figures and Tables

**Figure 1 nanomaterials-12-04133-f001:**
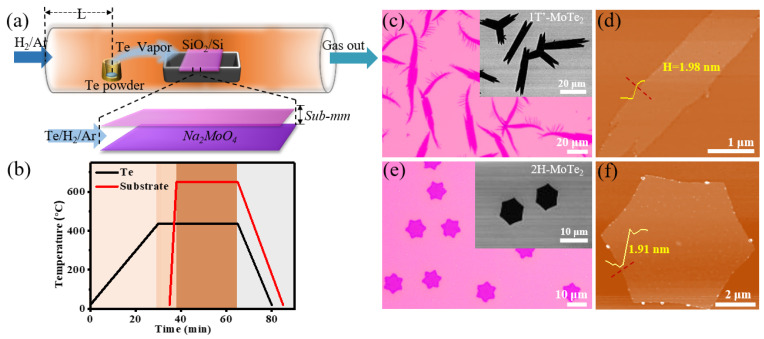
(**a**) Graphical representation of the space-confined CVD setup for the growth of atomically thin 2H-MoTe_2_ and 1T′-MoTe_2_; the upper one of the face-to-face substrates is bare while the bottom one is coated by Na_2_MoO_4_ solution. (**b**) Temperature profile for the Te powder (the black curve) and the face-to-face substrates (the red curve). (**c**,**d**) Optical microscopy image and atomic force microscopy (AFM) image of 1T′-MoTe_2_ grains grown on the top SiO_2_/Si substrate, respectively. The inset in (**c**) is the corresponding scanning electron micrograph (SEM) image. (**e**,**f**) Optical microscopy image and AFM image of 2H-MoTe_2_ grains grown on the upper SiO_2_/Si substrate, respectively. The inset of (**e**) is the corresponding SEM image.

**Figure 2 nanomaterials-12-04133-f002:**
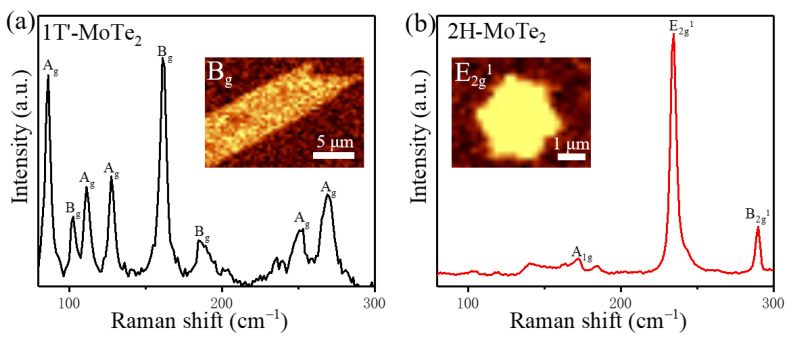
(**a**,**b**) Raman spectra of as-grown atomically thin 1T′-MoTe_2_ and 2H-MoTe_2_, respectively. The inset in (**a**) shows its Raman mapping of the B_g_ mode for 1T′-MoTe_2_. The inset in (**b**) shows Raman mapping of the E_2g_^1^ mode for 2H-MoTe_2_.

**Figure 3 nanomaterials-12-04133-f003:**
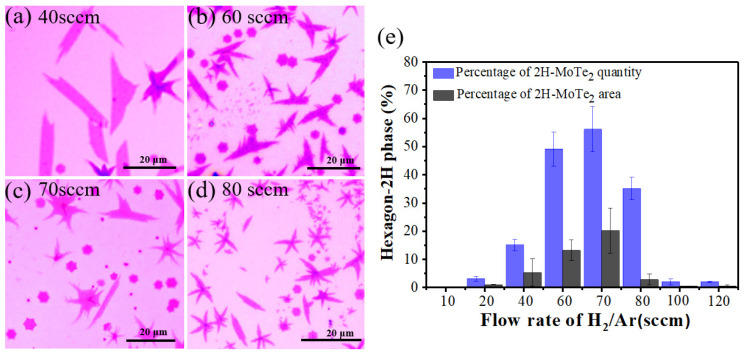
(**a**–**d**) Optical images of as-synthesized MoTe_2_ under different gas flow rates, 40, 60, 70 and 80 sccm, respectively. (**e**) Dependence of 2H-MoTe_2_ percentage in the quantity and area as a function of the gas flow rate which is positively correlated with the supply flux of Te.

**Figure 4 nanomaterials-12-04133-f004:**
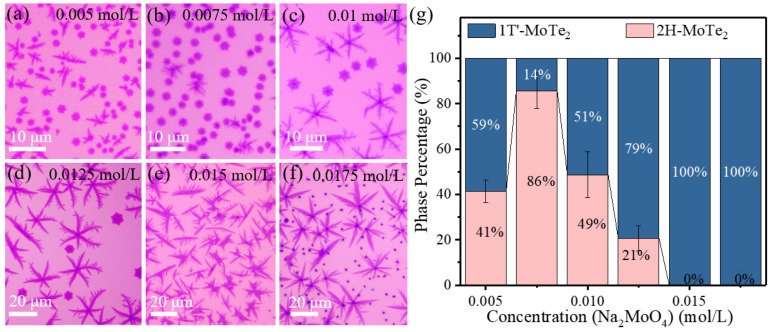
(**a**–**f**) Optical images of as-synthesized MoTe_2_ under different concentrations of the Na_2_MoO_4_ solution (0.005, 0.0075, 0.01, 0.0125, 0.015, 0.0175 mol/L) spin-coated on the bottom SiO_2_/Si substrate. (**g**) Dependence of 2H-MoTe_2_ and 1T′-MoTe_2_ percentage in the quantity as a function of the concentration of Na_2_MoO_4_ solution.

**Figure 5 nanomaterials-12-04133-f005:**
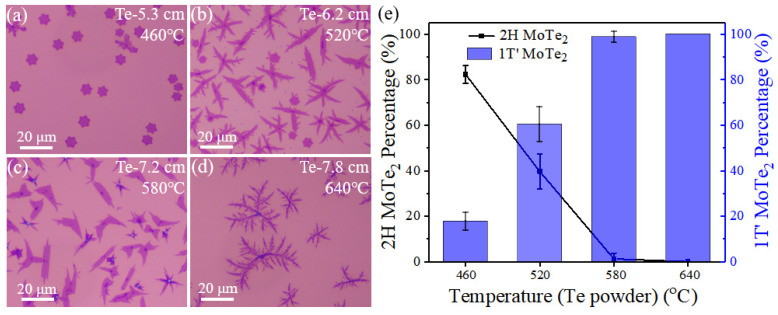
(**a**–**d**) Optical microscopic images of as-synthesized MoTe_2_ by varying the location of Te powder (L, the distance from the heating-zone edge in the upstream direction), related to different heated temperatures. (**e**) Dependence of the phase percentage for few-layer 2H and 1T′ MoTe_2_ as a function of the heating temperature of Te powder.

**Figure 6 nanomaterials-12-04133-f006:**
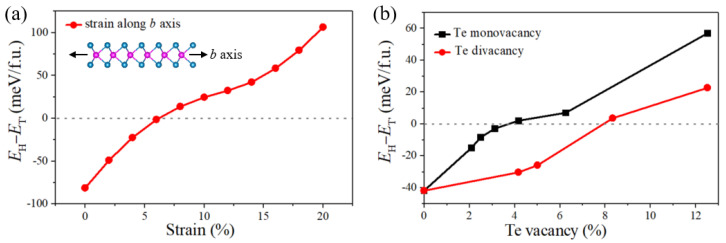
(**a**) The energy differences per formula between the 2H MoTe_2_ and 1T′ MoTe_2_ versus the strain along the *b* axis from the DFT calculation. (**b**) DFT calculation of the energy differences between the 2H and 1T′ MoTe_2_ per formula as a function of the Te monovacancy (black line) and Te divacancy concentration (red line).

**Figure 7 nanomaterials-12-04133-f007:**
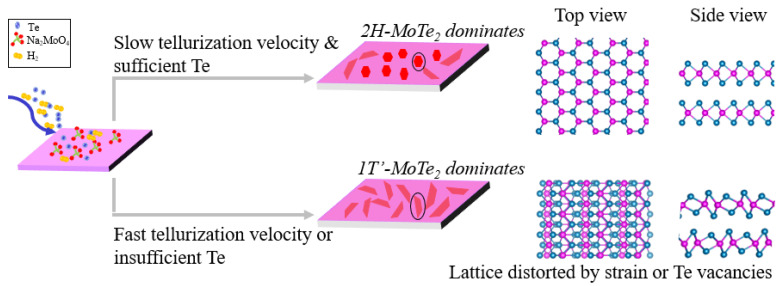
Schematic diagram of the growth mechanism for 2H-MoTe_2_ and 1T′-MoTe_2_.

## Data Availability

Not applicable.

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
