# Peer review of "Phase-Controllable Chemical Vapor Deposition Synthesis of Atomically Thin MoTe2"

_nanomaterials, 2022, doi:10.3390/nano12234133_

Round 1
Reviewer 1 Report
The manuscript presents Synthesis of 2D Atomically Thin MoTe2 films by using Chemical Vapor Deposition method. The authors varied the carrier gas flow rate, the concentration of Na2MoO4 solution and the heating temperature of Te powder to control the phase ratio from 2H to 1T’. The topic is very interesting, moreover the phase control is extremely challenging and attractive and provides a new method for phase-controllable synthesis of atomically thin TMDCs with semiconducting and metallic phases. Although the results are well presented in figures and diagrams, the paper is some parts is difficult to read and to understand the meaning -the sentences are too long, also with a lot of English grammar errors
I recommend the manuscript to be revised and discussions to be presented more clearly.
For example:
“Te powder was preheated for 5 minutes before the substrate with the Na2MoO4 precursor was pushed to the center of the furnace to ensure Te powder was sufficiently heated and diffused to the reaction zone as shown in the temperature profile of Figure 1b.”
“Considering the Te content plays an important role in the phase of the few-layer MoTe2 reported in the previous study [32,36,53,54], the gas flow rate which is closely related with the content of Te was investigated in our experiments.”
What means “proper tellurization velocity” -row 89?
Reviewer 2 Report
This is an interesting paper related to the experimental realization and theoretical study of two phases of 2D-MoTe2. The controllable synthesis of two phases (2H and 1T’) is reported. A theoretical understanding of the obtained experimental results is also provided based on DFT computations. There are basically two concerns regarding the manuscript: the English should be improved and the introduction should be complemented with references to some previous theoretical work done on 2D-MoTe2. The calculation details are also weak: the k-point greed is not provided and there is not clear whether any relaxation of atomic position (and also lattice constants) was done. The relaxation of atomic positions may be especially important if any point defects (e.g. vacancies) are included in the calculations.
Reviewer 3 Report
The subject matter of the manuscript “Phase-controllable chemical vapor deposition synthesis of atomically thin MoTe2” is extremely topical, as evidenced by the large number of similar publications in the last 8 years. According to WoS, the number of publications on MoTe2 thin films increased several times with a citation rate of 30 citations per work within 5 years. The work of Tao Xu et al. concerns the phenomenon of phase transition in thin films between the semiconductor and metallic state, very important for its possible application. The authors clearly presented the current knowledge in the subject of MoTe2 thin layer formation as well as the observed phase transition effect. The presented method of thin film preparation is proprietary and has already been published in the earlier works that are cited. The main topic of the work is the influence of internal stresses in a thin film resulting from excess or underflow of Te flux on the crystalline phase of MoTe2 islands. The observed results by microscopic methods and Raman spectroscopy were supported by ab initio calculations.
In my opinion, the work is extremely interesting and well prepared for publication. Nevertheless, I have some minor comments that I list below:
1. Despite the well-structured introduction, I have the impression that the authors did not fully refer to the latest publications. Cited in the manuscript is prior to 2015. By limiting my search to publications only on the phase transition in MoTe2 thin films alone, I found at least three of the last four years worth considering. (Nano Lett. 2019, 19, 6, 3612–3617, Nanoscale, 2018,10, 19964-19971, Extreme Mechanics Letters Volume 40, October 2020, 100946)
2. The presented theoretical model is not sufficiently described. How many unit cells in a layer are considered? What is the purpose of introducing a double layer if they are separated enough to not interact? The more so that in the experimental part the observations concern a 2ML islands. In addition, the thin films obtained in the experiment are formed on the SiO2 surface. Does the theoretical model take into account surface interaction?
3. Authors should check the correctness of cited papers. As an example, the paper cited under number [41] has no journal title.
If the authors correct the manuscript in accordance with the comments, the publisher should consider publishing it.
